# Profile of pro-inflammatory cytokines in colostrum of nursing mothers at the extremes of reproductive age

**Denise Vasconcelos de Jesus Ferrari[1], Jossimara Polettini[1,2], Lucas Lima de Moraes[3], Letícia Aguiar de Campos[3], Márcia Guimarães da Silva[4], Erika Kushikawa Saeki[5], Glilciane Morceli** [1,6]*

1 Mestrado em Ciências da Saúde-Universidade do Oeste Paulista/UNOESTE, Presidente Prudente, SP, Brasil, 2 Universidade Federal da Fronteira Sul/UFFS/Campus Passo Fundo, RS, Brasil, 3 Faculdade de Ciências da Saúde-Universidade do Oeste Paulista/UNOESTE, Presidente Prudente, SP, Brasil, 4 Faculdade de Medicina de Botucatu, Universidade Estadual Paulista/UNESP-Botucatu, SP, Brasil, 5 Instituto Adolfo Lutz-IAL/Presidente Prudente, SP, Brasil, 6 Universidade do Estado de Minas Gerais/UEMG/Campus Passos, MG, Brasil

* glilciane@gmail.com

**Data Availability Statement:** All relevant data are within the manuscript and its Supporting Information files. The Methodology of Elisa is

## Abstract

Gestations at the extremes of reproductive age are characterized as high-risk pregnancies, conditions that might influence colostrum composition. This first milk secretion contains nutrients necessary for the development and immunity of the newborn; therefore, this study aims to compare adolescent and advanced maternal age mothers regarding sociodemographic, gestational, and perinatal characteristics and the colostrum levels of pro-inflammatory cytokines in these groups of study. This cross-sectional study has compared sociodemographic, gestational and perinatal data from adolescent mothers (between 10 and 24 years old) (n = 117), advanced maternal age mothers (over 35 years of age) (n = 39) and mothers considered a control group (25 to 34 years old) (n = 58). Additionally, colostrum samples were obtained from the studied and control group subjects by manual milking, between 48 and 72 hours postpartum, and the samples were analyzed for cytokine concentrations by enzyme-linked immunosorbent assay (ELISA). The majority of the studied mothers reported living a stable union, and 81.2% of the adolescent mothers did not carry out any paid activity. Mothers with advanced maternal age mainly delivered by cesarean section and presented a higher body mass index (BMI). Neonatal weight and Apgar score were not different between the groups. The concentrations of interleukin (IL)-1β and IL-6 were higher in the colostrum of mothers with advanced age compared to adolescent mothers, but did not differ from the control group. The concentrations of IL-8 and tumor necrosis factor alpha did not differ between the three groups. Therefore, our data demonstrated that maternal age influenced the sociodemographic and gestational characteristics as well as the composition of colostrum cytokines.

described in the DOI: https://doi.org/10.17504/protocols.io.bdggi3tw.

**Funding:** There are no financial support.

**Competing interests:** The authors have declared that no competing interests exist.

## Introduction

Gestations at the extremes of reproductive age are have become more frequent in the last decade, and such conditions are characterized as high-risk pregnancies, defined as "those in which the life or health of the mother and/or the fetus and/or the newborn are more likely to be affected than the mean population considered" [1]. In this context, adolescence pregnancies present higher rates of obstetric complications, and are also considered a public health problem in some developing countries, as they carry social and biological outcomes, for example dropping out of school, social isolation, temporary or permanent education interruption, emotional instability and a stable union with a partner at an early stage of life [2,3,4].

Furthermore, the perinatal results from adolescent pregnant women can be adverse, mainly due to their biological immaturity regarding growth and development. Adversities include a greater number of low birth weight newborns (LBW, <2500g), higher risks of intrauterine growth restriction (IUGR), Apgar scores below seven in the fifth minute and, in particular, a higher preterm birth rate (<37 gestational weeks) [3].

On the other hand, pregnancies after the age of 35 years are considered late gestation, which has become increasingly frequent due to aspects such as birth control improvement, advances in assisted reproductive technology, late marriages, increased rates of divorce followed by new unions, women with a higher level of education, and advances in health care [5,6]. However, obstetric complications are observed in the pregnancies of advanced maternal age mothers, including antepartum hemorrhage, pregnancy-induced hypertension, diabetes, IUGR, anomalous presentation, macrosomia, dystocia, prolonged gestation, oligohydramnios, polyhydramnios, and preterm labor [5–7].

The main World Health Organization (WHO) strategy to reduce child mortality is the promotion of breastfeeding around the world [8]. Breastfeeding is considered an age-old practice, with numerous recognized nutritional, immunological, cognitive, economic, and social benefits [9]. Colostrum is the first produced milk secretion, which ordinarily occurs between the second and third postpartum day; the main colostrum components are cytokines, fats, immunoglobulins, proteins, carbohydrates, vitamins, leukocytes, lactoperoxidase, lactoferrin, and lysozyme, as well as hormones and growth-promoting peptides [10–12]. Among these components, the presence of some cytokines is suggested to compensate for the delay in development of the newborn immune system, although the specific role of cytokines is still controversial [13].

Cytokines are soluble glycoproteins that signal intercellular communication, and share several functions with each other. Their presence in human breast milk results in a cascade of effects involved in the regulation of inflammatory response and in the function of the newborn's immune system, such as their role in maturing the infant intestine [14]. The frequently studied cytokines in breast milk include interleukins (IL)-1β, IL-6, IL-8, IL-10 and IL-12 and tumor necrosis factor (TNF)-α, which are all found in the colostrum and mature milk secretion [13,15–17].

Primarily, IL-1β is an inflammatory mediator, involved in cell proliferation, differentiation, and apoptosis. The concentration of IL-1β in colostrum seems to be constant regardless of the timing of birth; however, in mature milk, its concentration is higher in preterm than in term deliveries. The action of IL-1β is not well known, but it appears to be involved in mammary gland defense mechanisms, including the production of immunoglobulin A and the cytokines, such as IL-6, IL-8, TNF-α, and IL-12 [15–17]. IL-6 is an important pro-inflammatory and anti-inflammatory cytokine produced by keratinocytes, fibroblasts, macrophages, endothelial cells, and T-cells. It is secreted by macrophages and T cells in response to trauma, such as burns or tissue inflammation, with a potent pyrogenic action. IL-6 stimulates the acute phase

response, B-cell differentiation, and antibody production by the same pathways in colostrum [12,17].

IL-8 is produced by phagocytes and epithelial cells and is capable of recruiting neutrophils. It is an important component of innate immunity and inflammatory response, has a physiological role in the development and maturation of the gut, and may be involved in the anti-infective protection mechanisms of breast milk. Another role of IL-8 in breast milk is related to newborn protection due to its strong chemotactic activity and activation of the cells involved in allergic diseases [18]. TNF-α is secreted by mammary epithelium and macrophages of breast milk, has a higher concentration in colostrum than in mature milk and participates in the mechanism of defense against infections and trauma [16–17].

Considering that maternal breast milk is the gold standard in feeding all newborns and that the effect of maternal age on breast milk composition is still poorly understood, the objectives of this study were to compare adolescent and advanced maternal age mothers regarding sociodemographic, gestational, and perinatal characteristics and the colostrum levels of pro-inflammatory cytokines in these groups.

## Materials and methods

A cross-sectional study was conducted with adolescent mothers and those of advanced maternal age and their newborns who attended the maternity unit of the Regional Hospital of Presidente Prudente, SP, Brazil, from March 2017 to July 2018. The work was approved by the Research Ethics Committee of the Universidade do Oeste Paulista, UNOESTE, Presidente Prudente, SP, Brazil, CAAE 67772617.8.0000.5515. The participants signed the informed consent form and the person responsible for the nursing mother under the age of 18 authorized the participation in the research by signing the Informed Consent Form and the Free and Informed Consent Form according to the Ethics in Research with Human Beings criteria as per Resolution no. 466/2012 of the National Health Council.

### Subjects

Nursing mothers and their newborns were recruited using the following inclusion criteria: gestational age at birth between 37 and 41 weeks and negative serological reactions for hepatitis, human immunodeficiency virus (HIV) and syphilis. Twin pregnancies, fetal malformations or loss of data regarding delivery and the neonatal period were the exclusion and/or discontinuity criteria.

Subjects were divided in two groups of study: adolescent mothers (between 10 and 24 years old) [2] and advanced maternal age (over 35 years of age). Mothers aged between 25 and 34 years old were considered the control group.

The weight classification of the newborns was small for gestational age (SGA), appropriate for gestational age (AGA), and large for gestational age (LGA) (weight/gestational age ratio) according to the service protocol.

### Colostrum collection

The colostrum was collected by manual expressing, always in the morning (from 8 to 10 am) and in the interval between two feedings, considering the period between 48 and 72 hours after delivery. Half of the samples from each study group and the control group were collected at 48 h postpartum and half at 72 h postpartum in order to avoid variations in the cytokine concentrations over time. A maximum colostrum volume of 10 mL was collected into sterile plastic tubes, immediately transported to the laboratory and then frozen and stored at -80˚C until analysis.

## Determination of IL-1β, IL-6, IL-8, and TNF-α in colostrum by ELISA assay

Specimens were thawed at 4°C, and a 2 mL aliquot of collected colostrum was centrifuged at 160 g for 10 minutes at 4°C. This allowed the separation of the sample into three phases: cell pellet, an intermediate aqueous phase, and the supernatant containing the fats, which was discarded; the aqueous supernatant was used for the dosage of the cytokines of interest [20]. Commercially available ELISA kits (R&D Systems, Minneapolis, MN, USA) were used to quantify IL-1β, IL-6, IL-8 and TNF-α levels in the studied samples. All assays were performed according to the manufacturer's instructions. Samples in which cytokine levels were estimated to be below the sensitivity of the assay were set as equal to the sensitivity of the assay and those with concentrations at levels above standard curve were diluted and re-assayed. The assays readers were performed in an ELISA automatic reader (Epoch-BioTek, Winooski, VT, USA), at a wavelength of 492 nm. The concentrations of cytokines in the colostrum were calculated on the standard curve obtained with different concentrations of the recombinant human cytokines of interest.

## Statistical analysis

Data on the sociodemographic and gestational variables were submitted to the $X^2$ or Fisher's exact test, or to the Kruskal-Wallis non-parametric test for comparison between the three study groups, after analysis of data normality by the Kolmogorov-Smirnov test. The concentration of cytokines in colostrum was analyzed by the Kruskal-Wallis test, followed by the Dunn test. Statistical analysis was performed using Graph Pad Prism software, version 6.0 (San Diego, CA), and the level of significance adopted for all tests was 5%.

# Results

## Sociodemographic and gestational characteristics of the mothers

The sample size was composed of 158 nursing mothers in the control group, 117 in the group of adolescents, and 39 in the advanced maternal age group. The reduced number in the latter group was justified by a smaller number of women who had recently given birth with advanced maternal age in the study period. Regarding pregnancy-related complications, 5 mothers in the adolescent group, 7 in the control group and 4 in the advanced maternal age group presented complications such as hypertension or gestational diabetes.

Table 1 depicts the sociodemographic characteristics of the studied mothers during the period considered. As expected, the median maternal age was statistically lower in the adolescent group ($p<0.0001$). The predominant marital status was the stable union in all groups ($p<0.0001$). The unpaid profession was predominant among the adolescent nursing group and paid profession in the advanced maternal age group ($p = 0.0002$). The variables ethnicity, smoking, alcohol consumption (up to 7 weekly doses), living with smokers, and practicing physical activity (3 or more times a week) did not differ between the groups ($p>0.05$).

Table 2 presents the gestational and obstetric characteristics of the nursing mothers included in the study. Body mass index (BMI) at the beginning and end of gestation was statistically lower in adolescents ($p<0.0001$); however, the weight gain was higher in this group ($p = 0.0004$). A higher number of multiparous women was observed in the advanced maternal age group ($p<0.0001$), which also presented predominant cesarean deliveries rates ($p = 0.0001$). The number of prenatal visits did not differ between groups ($p>0.05$).

## Perinatal outcomes of the newborns

The perinatal results of the nursing mothers' newborns are demonstrated in Table 3. The New Ballard perinatal characteristics birth weight, weight classification, Apgar score at 1 and 5

**Table 1. Sociodemographic characteristics of adolescent, control, and advanced maternal age nursing mothers included in the study.**

| Characteristics | Adolescents (n = 117) | Controls (n = 158) | AMA (n = 39) | p |
|---|---|---|---|---|
| *Maternal characteristics* | | | | |
| Age (years)[#] | 20 (12–23)[a0] | 29 (24–35)[b0] | 37 (36–46)[c0] | **<0.0001***  |
| Ethnicity [Ύ] | | | | |
| White | 50 (42.7) | 76 (48.1) | 12 (30.8) | 0.14 |
| Non White | 67 (57.3) | 82 (51.9) | 27 (69.2) | |
| Marital status [Ύ] | | | | |
| Single | 41 (35.0) | 17 (10.7) | 6 (15.4) | **<0.0001***  |
| Stable union | 76 (65.0) | 141 (89.3) | 33 (84.6) | |
| Profession [Ύ] | | | | |
| Paid | 22 (18.8) | 62 (39.3) | 21 (53.8) | **0.0002***  |
| Unpaid | 95 (81.2) | 96 (60.7) | 18 (46.2) | |
| Smoking habit [Ύ] | | | | |
| Yes | 6 (5.1) | 16 (10.1) | 7 (17.9) | 0.13 |
| Ex-smoker | 18 (15.4) | 17 (10.7) | 4 (10.2) | |
| Mother living with smokers [Ύ] | | | | |
| Yes | 45 (38.4) | 48 (30.4) | 14 (35.9) | 0.36 |
| Consumed Alcohol [Ύ] | | | | |
| Yes | 3 (2.6) | 1 (0.6) | 1 (2.6) | 0.49 |
| Physical activity practice [Ύ] | | | | |
| Yes | 11 (9.4) | 8 (5.1) | 2 (5.1) | 0.64 |

[#] Kruskal-Wallis test, followed by Dunn test for comparison between groups: median (minimum—maximum);

[Ύ] $X^2$ Test or Fisher's exact test, n (%);

[*] statistically significant ($p<0.05$); In lines, values followed by the same letters and the same index do not statistically differ.

**Table 2. Gestational and obstetric characteristics of adolescent, control and advanced maternal age nursing mothers included in the study.**

| Characteristics | Adolescents (n = 117) | Controls (n = 158) | AMA (n = 39) | p |
|---|---|---|---|---|
| *Gestational and Obstetric Characteristics* | | | | |
| Body mass index (BMI) [#] | | | | |
| At the beginning of gestation | 21.8 (14.3–38.2) [a1] | 25.8 (16.4–49.9) [b1] | 28.7 (19.5–43.0) [b1] | **<0.0001***  |
| At the end of gestation | 26.6 (19.2–40.50)[a2] | 30.0 (20.4–63.9)[b2] | 31.4 (23.3–51.4) [b2] | **<0.0001***  |
| Weight gain (kg)[#] | 11.9 (-4.3–23.3) [a3] | 9.5 (-6.3–70.0) [b3] | 7.0 (-2.0–32.0) [b3] | **0.0004***  |
| Parity [Ύ] | | | | |
| Primigravida | 80 (68.4) | 32 (20.3) | 2 (5.1) | **<0.0001***  |
| Secundigravida | 26 (22.2) | 50 (31.6) | 6 (15.4) | |
| Multigravida | 11 (9.4) | 76 (48.1) | 31 (79.5) | |
| Number of prenatal consultations [Ύ] | | | | |
| ≤7 consultations | 30 (25.6) | 47 (29.7) | 12 (30.8) | 0.70 |
| > 7 consultations | 87 (74.4) | 111 (70.3) | 27 (69.2) | |
| Type of birth [Ύ] | | | | |
| Vaginal | 81 (69.2) | 97 (61.4) | 12 (30.8) | **0.0001***  |
| Cesarean | 36 (30.8) | 61 (38.6) | 27 (69.2) | |

[#] Kruskal-Wallis test, followed by Dunn test for comparison between groups: median (minimum—maximum);

[Ύ] $X^2$ Test or Fisher's exact test, n (%);

[*] statistically significant ($p<0.05$); In lines, values followed by the same letters and the same index do not statistically differ.

**Table 3. Perinatal results of newborns of adolescent, control, and advanced maternal age nursing mothers included in the study.**

| Characteristics | | Adolescents (n = 117) | Controls (n = 158) | AMA (n = 39) | p[*] |
|---|---|---|---|---|---|
| New Ballard (weeks, days) [#] | | 39w (35w1d – 41w) | 38w3d (34w2d – 41w) | 38w3d (37w – 40w3d) | 0.18 |
| Birth weight (Kg) [¥] | | 3.252 ± 0.431 | 3.284 ± 0.483 | 3.297 ± 0.443 | 0.80 |
| Weight classification [Ɣ] | | | | | |
| | PIG | 7 (6.0) | 11 (7.0) | 2 (5.1) | 0.58 |
| | AIG | 104 (88.9) | 136 (86.0) | 32 (82.1) | |
| | GIG | 6 (5.1) | 11 (7.0) | 5 (12.8) | |
| Apgar at 1 minute [Ɣ] | | | | | |
| | ≥7 | 110 (94.0) | 154 (97.5) | 35 (89.7) | 0.09 |
| | <7 | 7 (6.0) | 4 (2.5) | 4 (10.3) | |
| Apgar at 5 minutes [Ɣ] | | | | | |
| | ≥7 | 115 (98.3) | 158 (100.0) | 39 (100.0) | 0.18 |
| | <7 | 2 (1.7) | 0 | 0 | |
| Days of hospitalization [#] | | 2 (2–6) | 2 (2–8) | 2 (2–3) | 0.09 |

[#] Kruskal-Wallis test, followed by Dunn test for comparison among groups: median (minimum—maximum);

[¥] ANOVA test, followed by comparative test, mean ± SD;

[Ɣ] $x^2$ test or Fisher's exact test, n (%);

[*] statistically significant ($p < 0.05$).

minutes of life, and days of hospitalization were analyzed. The perinatal outcomes of the nursing mothers at the extremes of reproductive ages were not statistically different.

## Colostrum cytokine concentrations

The sample size used for cytokine concentration analysis was 40 in the control group and adolescents group, and 39 in the group of mothers with advanced maternal age, as this was the minimal number of samples needed to detected differences in cytokine levels. Intra-assay coefficient of variation, detection limit and median values of cytokine detection are presented in Table 4. Approximately 10% of the samples presented levels below the limit of detection and the levels of IL-1β and IL-6 were significantly increased in colostrum samples from the advanced age mothers compared to adolescent mothers ($p < 0.05$), but did not differ in relation to the control group. The concentrations of IL-8 and TNF-α did not present differences between the three studied groups (Table 4).

## Discussion

The present study has compared the sociodemographic, gestational, and perinatal characteristics, as well as the colostrum cytokine concentrations from nursing mothers at the extremes of reproductive age with full-term gestation.

**Table 4. Concentration of the evaluated cytokines in colostrum of adolescent, control and advanced maternal age nursing mothers included in the study.**

| Cytokine (pg/mL) [#] | Adolescents (n = 40) | Control (n = 40) | AMA (n = 39) | p |
|---|---|---|---|---|
| IL-1β | 18.99 (2.37–96.55) [a4] | 26.28 (0.99–261.0) [a4,b4] | 31.32 (7.15–250.0) [b4] | **0.019**[*] |
| IL-6 | 20.73 (2.68–90.87) [a5] | 24.94 (1.62–97.99) [a5,b5] | 42.82 (0.17–195.5) [b5] | **0.04**[*] |
| IL-8 | 1276.0 (51.71–1921.0) | 1621.0 (70.6–1927.0) | 1602.0 (484.4–2080.0) | 0.14 |
| TNF-α | 79.97 (14.79–2600.0) | 84.76 (3.66–3369.0) | 109.8 (22.79–5113.0) | 0.22 |

[#]Kruskal-Wallis test, followed by Dunn test for comparison between groups: median (minimum—maximum). In lines, values followed by the same letters and the same index do not differ;

[*] statistically significant ($p < 0.05$).

Brazilian demographic characteristics are heterogeneous and, although changes in family structures currently trend towards an increase in single parents, mainly single women, our findings were different as they showed the predominance of a stable union in both groups. This might reflect the regional habits regarding the preference of couples to form a stable union by dividing responsibilities in relation to financial expenses and child care, as observed in 65–70% of the Brazilian population [19,20]. When analyzing the variable profession, it was observed that unpaid activity prevailed in the adolescent group, while paid activity was predominant in the advanced maternal age group. We suggest that the results of our study may refer to the demands of the labor market, with a preference for professionals with experience and/or to the lack of interest that adolescents demonstrate in finding a job.

Parity tends to increase with the age, as described in a multicenter study involving Brazilian population [19]. The authors reported that women between the ages of 15 and 19 have 0.18 children per woman on average, while those at the end of their reproductive period (between the ages of 45 and 49) have a mean parity of 3.14. In our study, although the number of children was not evaluated, we similarly observed almost 80% of mothers over 35 years presenting three or more pregnancies. However, the mode of delivery was predominantly cesarean section in the advanced age women compared to adolescents; previous data suggest that maternal conditions linked to fetal impairment are more frequent and more severe in this age group [5,6,21]. In this context, Domingues et al. [22] reported the reasons for cesarean deliveries in Brazilian hospitals, including women with complications in pregnancy, which, in turn, is more frequent in advanced age women. Besides an occasional pregnancy disease or complication, the authors showed that the main motivation for cesarean choice was the fear of pain, which increases this mode of delivery in Brazilian hospitals, regardless of the diagnosis of complications or the sector (private or public) in which women work.

Several factors may influence the composition of maternal milk, such as demographic data, age, ethnicity, and gestational characteristics, as well as BMI, parity, and gestational period [23, 24]. Sinanoglou et al. [24] evaluated the possible impact of sociodemographic factors in colostrum composition in Greek nursing mothers, and found that the fatty acid profile and colostrum fat mainly depend on nationality and age rather than on the mode of delivery or maternal pregnancy BMI. In the present study, we found higher BMI in the advanced maternal age group, which is higher than the mean from postpartum Brazilian women (23.6±4.6), as described by de Castro et al. [20]. Conversely, these authors demonstrated that women of older age were more likely to adhere to a healthy dietary pattern, which suggests a regional disparity; however, diet was not evaluated in this study.

It is important to mention that prematurity has been implicated as a factor influencing the cytokine concentration in breast milk. Trend et al. [25] showed that maternal milk from mothers who are breastfeeding preterm newborns contains significantly higher concentrations of certain immune system factors (TGF-β2 and antimicrobials, such as beta-defensin 1) compared to breast milk from full-term newborns. Moreover, Gila-Diaz et al. [26] reviewed the literature and reported that the highest levels of bioactive factors, including cytokines, are found in the colostrum; these decrease throughout the lactation period, being found at higher levels in preterm compared to full-term milk [25]. However, these studies compared gestational age at birth categorized as preterm and term and did not consider adolescent and advanced maternal age ranges.

Breast milk has a protective role against neonatal sepsis, probably mediated by antimicrobial components and modulation of the neonatal immune system. Chemokines in human milk, such as IL-8, and their receptors CXC chemokine receptors (CXCR)-1 and CXCR-2, can activate the immune system of the neonate against invading pathogens [18,26,27]. In our study, when comparing the concentration of IL-8 in colostrum at the extremes of maternal

age, we observed that there were no significant differences between the nursing mothers. Similarly, previous data showed no differences regarding IL-8 levels between term and preterm mother's milk [27]. Additionally, our results may reflect those from Munblit et al. [13], who reported that cytokines are present at much lower levels than growth factors in breast milk, and that the cytokine decline over time was less consistent, suggesting that the role of IL-8 is constant during breastfeeding.

Some obstetric conditions may also influence cytokine levels, and although these were not addressed in the present study, they are important for understanding the dynamics of these mediators found in biological fluids. When considering adverse conditions such as preeclampsia, Freitas et al. [28] showed that this condition alters the levels of pro-inflammatory cytokines in breast milk, given that there is an increase in IL-1β and IL-6 levels in the colostrum of mothers with preeclampsia. Although the study did not consider reproductive age, the risk group for preeclampsia was maternal age over 35 years [29], which may also influence cytokines. This was observed in our study, which demonstrated higher IL-1β and IL-6 concentrations in the colostrum from advanced maternal age mothers compared to adolescent mothers. In preeclampsia, higher levels of cytokines in human milk are related to immunological signals for the host defense in high-risk neonates [30]. Therefore, the increased levels of IL-1β and IL-6 in the colostrum of mothers with advanced maternal age, evidenced in the present study, may be beneficial to the newborn.

Therefore, our findings have added knowledge regarding the colostrum composition of cytokines at the extremes of reproductive age, as there is a scarcity of data to date. Maternal age influenced colostrum composition, likely due to the biological immaturity of adolescent mothers. However, further studies should be carried out to better understand the inflammatory environment in pregnancy, to provide additional incentives for breastfeeding and avoid premature weaning and to improve the quality of life for both mothers and their children, as well as stimulating the mother/newborn bond.

## Limitations of the study

There are some limitations to the present study that may have influenced the results or understanding and analysis of the results, such as the absence of concentrations of cytokines in the milk during the different periods of lactation, the lack of evaluation of the presence or absence of infection and allergic diseases in nursing mothers, and, finally, our analysis covered only a few cytokines, so the results may not be generalizable for all immunological factors.

## Acknowledgments

The authors would like to thank the Universidade do Oeste Paulista, together with the Regional Hospital of Presidente—SP, which provided the structures for this work to be carried out. In addition, we would like to thank the Adolf Lutz Institute and the Laboratory of Immunopathology of the Maternal-Fetal Relationship—UNESP Botucatu-SP, where the analyses were performed.

## Author Contributions

**Conceptualization:** Denise Vasconcelos de Jesus Ferrari, Jossimara Polettini, Lucas Lima de Moraes, Letícia Aguiar de Campos, Glilciane Morceli.

**Formal analysis:** Jossimara Polettini, Márcia Guimarães da Silva, Glilciane Morceli.

**Investigation:** Lucas Lima de Moraes, Letícia Aguiar de Campos, Erika Kushikawa Saeki, Glilciane Morceli.

**Methodology:** Denise Vasconcelos de Jesus Ferrari, Jossimara Polettini, Lucas Lima de Moraes, Letícia Aguiar de Campos, Erika Kushikawa Saeki, Glilciane Morceli.

**Project administration:** Glilciane Morceli.

**Supervision:** Glilciane Morceli.

**Writing – original draft:** Denise Vasconcelos de Jesus Ferrari, Glilciane Morceli.

**Writing – review & editing:** Jossimara Polettini, Márcia Guimarães da Silva, Erika Kushikawa Saeki, Glilciane Morceli.

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
