## [Decision Letter · Decision Letter 0]

28 Jan 2020

PONE-D-19-35580

Profile of inflammatory cytokines in colostrum of nursing mothers at the extremes of reproductive age. A cross-sectional study.

PLOS ONE

Dear Dr Morceli,

Thank you for submitting your manuscript to PLOS ONE. After careful consideration, we feel that it has merit but does not fully meet PLOS ONE’s publication criteria as it currently stands. Therefore, we invite you to submit a revised version of the manuscript that addresses ALL of the points raised during the review process.

We would appreciate receiving your revised manuscript by Mar 13 2020 11:59PM. To enhance the reproducibility of your results, we recommend that if applicable you deposit your laboratory protocols in protocols.io, where a protocol can be assigned its own identifier (DOI) such that it can be cited independently in the future. For instructions see: http://journals.plos.org/plosone/s/submission-guidelines#loc-laboratory-protocols

We look forward to receiving your revised manuscript.

Kind regards,

Frank T. Spradley

Academic Editor

PLOS ONE

Journal Requirements:

Reviewers' comments:

Reviewer's Responses to Questions

**Comments to the Author**

1. Is the manuscript technically sound, and do the data support the conclusions?

Reviewer #1: Partly

Reviewer #2: Yes

Reviewer #3: No

Reviewer #4: Yes

2. Has the statistical analysis been performed appropriately and rigorously? 

Reviewer #1: No

Reviewer #2: Yes

Reviewer #3: N/A

Reviewer #4: Yes

3. Have the authors made all data underlying the findings in their manuscript fully available?

Reviewer #1: Yes

Reviewer #2: Yes

Reviewer #3: Yes

Reviewer #4: Yes

4. Is the manuscript presented in an intelligible fashion and written in standard English?

Reviewer #1: No

Reviewer #2: No

Reviewer #3: Yes

Reviewer #4: Yes

5. Review Comments to the Author

Reviewer #1: Although the examination of the profile of inflammatory cytokines in colostrum of nursing mothers at the extremes of reproductive age seems to be very interesting, a number of points need clarifying and certain statements require further justification. These are given below.

<points>

1. In this manuscript, the authors measured concentrations of IL-1β, IL-6, IL-8 and TNF-α in colostrum of adolescent, control, and advanced maternal age nursing mothers (Table 4). The numbers of adolescents, controls, and advanced maternal age nursing mothers were 40, 40, and 39, respectively. In contrast, the numbers of socio-demographic characteristics (Table 1), gestational and obstetric characteristics (Table 2), and perinatal results of newborns (Table 3) of adolescents, controls, and advanced maternal age nursing mothers were 117, 158, and 39, respectively. They (adolescent and control) are different groups. Therefore, the tables 1-3 should be changed, including 40 adolescents and 40 controls that are identical to table 4.

2. How ELISA were performed especially what ELISA kits were used should be described in “Materials and methods”.

4. In references, Refs. 1-4, 6-7, 9, 11, 13, 17, and 23 were not English references. Most of readers are unable to understand whether they are reasonable/appropriate. The authors should change the references to English references.

5. The manuscript contains multiple errors in English writing (e.g. lines 33, 59, 105, 132, 134, 147, 165, etc.); therefore, it should be checked by native English speakers/scientists before submission.

Reviewer #2: Review: Manuscript number: PONE-D-19-35580

General comments:

The manuscript describes the profile of cytokines in mothers at early (10-24 years) vis-à-vis those at late gestational age (≥35 years) relative to a control group (25-34).The study is of interest considering that both populations are within the high risk group for complications for mother and infant. Colostrum is notably for defense and protection of the baby, rather than as a nutrient so assessment of cytokine profiles is relevant.

The article can be improved with some help in the style of writing and in the grammar. Details on specific sections are presented below.

Title: The title could benefit from a minor modification: cytokines are either pro- or anti-inflammatory or both so “selected” could replace “inflammatory” in the title.

Keywords: I suggest you replace “extremes age” with “ age extremes”

Abstract:

Line 39: the age classification is incorrect in the abstract

Methods: The methods section of the abstract needs significant editing to improve the sentence structure and grammar. The results section also need to be improved.

Introduction:

Line 89: …glycoproteins that signal…….

Line 92: ….and function of the newborn’s immune……………

Line 105 : reference numbering has an error

Line 106-107: Please improve the sentence.

Methods

Lines 124-125: Please improve the sentence

Line 133-137: Please revise

Lines 138-139: The sentence is too long. Break it up appropriately.

Lines 142-147: Please revise

Lines 151-152: The dependent variables are not determination of the concentrations……..

Lines 154-155: Please write out the full meaning of the abbreviated letters

Lines 159 -163: Please be specific on the exact times when samples were collected. What time in the morning? How many samples were collected within 48 hours and how many within 72 hours. These are important because there are established variations in the cytokine concentrations with time.

Line 156: This should be under the results section

Discussion

Line 232-233: Please modify

Line 281-286: IL-6 is proinflammatory in the acute phase and involved in immune regulation of the inflammatory response. Its secretion is augmented by TNF-α and IL-1 β so it is not surprising that both are elevated although TNF-α is not. Considering that these are elevated in pre-eclampsia and sickle cell disease in pregnancy, it seems that the role of IL-1 β in augment IL-6 secretion in pregnancy is highly significant compared to TNF-α. Incidentally the risk group for preeclampsia is maternal age >35 years. This is quite interesting in understanding the inflammatory environment in pregnancy.

Conclusion:

The data is relevant and the article is publishable after the corrections are made.

Reviewer #3: The manuscript titled “Profile of inflammatory cytokines in colostrum of nursing mothers at the extremes of reproductive age” investigated the effect of maternal reproductive age (including adolescents group, control group (25-34 years), advanced maternal age group) on socio-demographic, gestational and perinatal parameters and in particular on the composition of pro-inflammatory cytokines in colostrum. Although it is of interest how the maternal reproductive age influences the different parameters that were investigated within this study, the reviewer does not understand the rationale behind this investigation.

Already the abstract should clearly point out why the authors aimed to study the composition of pro-inflammatory cytokines in the colostrum of the different groups. A first explanation is provided in the last paragraph of the introduction section.

Following several main questions remain unanswered:

What was the reason to present data on the marital status and profession of the study subjects? Was there any association expected to the composition of the colostrum cytokines? The reviewer would rather expect differences in the composition of the colostrum cytokines if there were any pregnancy-related complications (infections, preeclampsia, IUGR, ect) caused by the age of the mother; however this was either not investigated or there were no differences between the groups. It would be helpful to include analyses that are more detailed.

It would be of interest to analyze the socio-demographic, gestational and perinatal parameters only in those study subjects where cytokine data are available.

Do the authors have any information about the development of the neonates, for instance the infection rate during the first year of life? This could be correlated to the composition of the colostrum cytokines.

Very often, it is not clear between which groups a significant difference was detected.

Did the authors include those samples where cytokines could not be detected at all in their analysis?

What can the authors tell about changes in the colostrum cytokine composition within the first 72h after birth? These changes may influence their results as samples were taken between 48 and 72h after delivery.

Reviewer #4: The current study compared the concentrations of IL-1β, IL-6, IL-8, and TNF-α cytokines in the colostrum of adolescent mothers and those with advanced maternal age and to describe sociodemographic, gestational, and perinatal characteristics. The large sample size and cross sectional nature of the study is a significant strength of the study. The main findings of the study are that concentrations of IL-1β and IL-6 were higher in the colostrum of mothers with advanced age compared to adolescent mothers, but did not differ from the control group. Although the findings are modest, the study is carefully conducted, dataset is rich with regards to prenatal and postnatal observations in the mothers and offspring, limitations are acknowledged and the manuscript is well written. I have some minor comments that need to be addressed below.

• What ELISA platform (multiplex, single plex, company) was used for the cytokine analysis?

  </points>

6. PLOS authors have the option to publish the peer review history of their article (what does this mean?). If published, this will include your full peer review and any attached files.

Reviewer #1: No

Reviewer #2: Yes: Isaac K. Quaye

Reviewer #3: No

Reviewer #4: No

---

## [Author Response · Author response to Decision Letter 0]

17 Mar 2020

Review: Manuscript number PONE-D-19-35580

Response to reviewers

All answers are detailed below each reviewers’ comments, as well highlighted in the revised version of the manuscript. 

REVIEW COMMENTS TO THE AUTHOR

Reviewer #1: Although the examination of the profile of inflammatory cytokines in colostrum of nursing mothers at the extremes of reproductive age seems to be very interesting, a number of points need clarifying and certain statements require further justification. These are given below.

1) In this manuscript, the authors measured concentrations of IL-1β, IL-6, IL-8 and TNF-α in colostrum of adolescent, control, and advanced maternal age nursing mothers (Table 4). The numbers of adolescents, controls, and advanced maternal age nursing mothers were 40, 40, and 39, respectively. In contrast, the numbers of socio-demographic characteristics (Table 1), gestational and obstetric characteristics (Table 2), and perinatal results of newborns (Table 3) of adolescents, controls, and advanced maternal age nursing mothers were 117, 158, and 39, respectively. They (adolescent and control) are different groups. Therefore, the tables 1-3 should be changed, including 40 adolescents and 40 controls that are identical to table 4.

Answer: We thanks the reviewer for this observation, and we justify that a higher number of sociodemographic data was included in order to achieve one of our objectives regarding the comparison between adolescent and advanced maternal age mothers’ characteristics, besides cytokines colostrum levels. We’ve re-written the abstract and the final paragraph of introduction to make this purpose clearer. We agree the matched numbers would be ideal, however, 40 samples were the minimal number of samples needed to detected cytokines level differences. Therefore, due to the high cost of the technique, we decided to maintain the descriptive data from all studied patients in order to characterize the regional population during the studied period, and perform the Elisa assays in the minimal number of samples. Moreover, we took care of collecting half of the samples in the period of 48h postpartum, and half of them after 72h, which difficult increasing the number of colostrum collected in each group (please see answer to reviewer #2, question #7). In order to reply reviewer’s concern about the number of samples, we performed a statistical analysis for sociodemographic data considering only the same samples tested for cytokines, and we’ve obtained the same trend of results, therefore, the population characteristics are better represented by a larger number of samples, which also represents the reality in the Service during the studied period. 

2. How ELISA were performed especially what ELISA kits were used should be described in “Materials and methods”.

Answer: We provided the detailed description of the assays in the revised version of the manuscript (highlighted in blue), as well as the technique protocol was sent for assigned its own identifier (DOI), as suggested by Plos One. 

DOI number: dx.doi.org/10.17504/protocols.io.bdggi3tw

Lines 158-165: “Commercially available ELISA kits (R&D Systems, Minneapolis, MN, USA) were used to quantify individually the IL-1β, IL-6, IL-8 e TNF-α levels in the studied samples. All assays were performed according to the manufacturer’s instructions. Samples in which cytokine levels were estimated to be below the sensitivity of the assay were set equal to the sensitivity of the assay and those with concentrations at levels above standard curve were diluted and re-assayed. The assays readers were performed in an ELISA automatic reader (Epoch-BioTek, Winooski, VT, USA), at a wavelength of 492 nm.”

3. In references, Refs. 1-4, 6-7, 9, 11, 13, 17, and 23 were not English references. Most of readers are unable to understand whether they are reasonable/appropriate. The authors should change the references to English references.

Answer: We appreciate this cautious observation and we clarify all references were reviewed and changed to English references, and replaced when appropriate.

4. The manuscript contains multiple errors in English writing (e.g. lines 33, 59, 105, 132, 134, 147, 165, etc.); therefore, it should be checked by native English speakers/scientists before submission.

Answer: We clarify the text has been professionally proofread for improving writing and grammar (please find attached the certificate at the end of this letter).

Reviewer #2: 

General comments:

The manuscript describes the profile of cytokines in mothers at early (10-24 years) vis-à-vis those at late gestational age (≥35 years) relative to a control group (25-34).The study is of interest considering that both populations are within the high risk group for complications for mother and infant. Colostrum is notably for defense and protection of the baby, rather than as a nutrient so assessment of cytokine profiles is relevant.

The article can be improved with some help in the style of writing and in the grammar. Details on specific sections are presented below.

Answer: We thank your review and we clarify the text has been professionally proofread for improving writing and grammar (please find attached the certificate at the end of this letter).

1) Title: The title could benefit from a minor modification: cytokines are either pro- or anti-inflammatory or both so “selected” could replace “inflammatory” in the title.

Answer: We´ve agreed with reviewer´s suggestion and the change to “pro-inflammatory cytokines” was done in the title, as highlighted in blue in the revised version of the manuscript. 

2) Keywords: I suggest you replace “extremes age” with “ age extremes”

Answer: The suggested change was performed. 

3) Abstract:

Line 39: the age classification is incorrect in the abstract

Answer: Thanks for the observation, we made the description of adolescent mother consistent with material and methods section content. We’ve considered the adolescence as 10-24 years following the description by Sawyer et al. (1), as the studied population present similar characteristics regarding financial and emotional dependence in this period. We’ve included this reference in the manuscript (ref.#2).

(1) Sawyer S, Azzopardi P, Wickremarathne D, Patton G. The age of adolescence. The Lancet Child & Adolescent Health 2018;2:223–8.

4) Methods: The methods section of the abstract needs significant editing to improve the sentence structure and grammar. The results section also need to be improved.

Answer: The suggested changes were performed, as well as the structure and grammar were reviewed by a professional service as clarified above.

5) Introduction:

Answer: We appreciate your careful review, and the suggested changes were performed, as highlighted in blue in the revised version of the manuscript in the respective lines named below. Line 89: …glycoproteins that signal……. (line 89)

Line 92: ….and function of the newborn’s immune…………… (line 91-92)

Line 105 : reference numbering has an error (line 106)

Line 106-107: Please improve the sentence. (lines 107-108)

6) Methods

Answer: We appreciate your careful review, and the suggested changes were performed, as highlighted in blue in the revised version of the manuscript. We have expanded the abbreviations as suggested.

Lines 124-125: Please improve the sentence (lines 123-125)

Line 133-137: Please revise (lines 135-138)

Lines 138-139: The sentence is too long. Break it up appropriately.

The suggested changes were performed.

Lines 142-147: Please revise

The information regarding total sample size was removed under results section, as we’ve characterized the population considering the period of the study (please also see the answer to question #1, reviewer #1)

Lines 151-152: The dependent variables are not determination of the concentrations.

The suggested changes were performed.

Lines 154-155: Please write out the full meaning of the abbreviated letters (lines 142-143)

7) Lines 159 -163: Please be specific on the exact times when samples were collected. What time in the morning? How many samples were collected within 48 hours and how many within 72 hours. These are important because there are established variations in the cytokine concentrations with time.

Answer: We thank the reviewer for this important observation, and a better description was included in Methods section (lines 146-152).

We clarify that we took care of collecting half of the samples in the period of 48h postpartum, and half of them after 72h, which difficult increasing the number of colostrum collected in each group.

8) Line 156: This should be under the results section

Answer: The suggested changes were performed, as highlighted in blue in the revised version of the manuscript (lines 182-185).

9) Discussion

Line 232-233: Please modify

Answer: The suggested changes were performed, as highlighted in blue in the revised version of the manuscript (lines 231-232)

10) IL-6 is proinflammatory in the acute phase and involved in immune regulation of the inflammatory response. Its secretion is augmented by TNF-α and IL-1 β so it is not surprising that both are elevated although TNF-α is not. Considering that these are elevated in pre-eclampsia and sickle cell disease in pregnancy, it seems that the role of IL-1 β in augment IL-6 secretion in pregnancy is highly significant compared to TNF-α. Incidentally the risk group for preeclampsia is maternal age >35 years. This is quite interesting in understanding the inflammatory environment in pregnancy.

Answer: We thanks the reviewer for this interesting correlation, thus we added some discussion regarding cytokines dynamics in order to improve the interpretation of our data (lines 274-287). 

Conclusion:

The data is relevant and the article is publishable after the corrections are made.

Reviewer #3: The manuscript titled “Profile of inflammatory cytokines in colostrum of nursing mothers at the extremes of reproductive age” investigated the effect of maternal reproductive age (including adolescents group, control group (25-34 years), advanced maternal age group) on socio-demographic, gestational and perinatal parameters and in particular on the composition of pro-inflammatory cytokines in colostrum. Although it is of interest how the maternal reproductive age influences the different parameters that were investigated within this study, the reviewer does not understand the rationale behind this investigation. Already the abstract should clearly point out why the authors aimed to study the composition of pro-inflammatory cytokines in the colostrum of the different groups. A first explanation is provided in the last paragraph of the introduction section. 

Answer: We agree with the reviewer, and the abstract was re-written for a better consistence between the context and the objective of this study. We hope we’ve clarified the reasonability of our work:

Abstract: Gestations at the extremes of reproductive age are characterized as high-risk pregnancies, conditions that might influence colostrum composition. This first milk secretion contains the nutrients necessary for the development and immunity of the newborn, therefore, this study aims to compare adolescent and advanced maternal age mothers regarding sociodemographic, gestational, and perinatal characteristics and regarding colostrum levels of pro-inflammatory cytokines in these groups of study.

1) Following several main questions remain unanswered: What was the reason to present data on the marital status and profession of the study subjects? 

Answer: As mentioned in the previous reviewer’s question, one of the objectives of this study was to characterize the population regarding sociodemographic, gestational, and perinatal characteristics, as the such data are scarce considering the extremes of reproductive age population.

2) Was there any association expected to the composition of the colostrum cytokines? 

Answer: Considering the scarcity of data, the cytokine concentration may reflect either immaturity of adolescent mothers or a compensation for high-risk in advanced maternal age group. Considering other complications that affects this last group, we suggest increase of colostrum cytokines may be beneficial to the newborn, and some discussion regarding this point was added to the text (please see lines 274-287). 

3) The reviewer would rather expect differences in the composition of the colostrum cytokines if there were any pregnancy-related complications (infections, preeclampsia, IUGR, ect) caused by the age of the mother; however this was either not investigated or there were no differences between the groups. It would be helpful to include analyses that are more detailed. 

Answer: That’s an important point, we specify here that 5 mothers in adolescent group, 7 in control group and 4 in advanced maternal age group presented complications as hypertension or gestational diabetes (others were not analyzed). Although the studied groups are high-risk ones, we considered the low number of pregnancy-related complications are due to the inclusion of term births, as such complications would likely lead to preterm birth. In this view, we did not perform a secondary analysis of cytokines levels as number of samples in complicated groups is too small. We included this information in Results section (lines 182-185).

4) It would be of interest to analyze the socio-demographic, gestational and perinatal parameters only in those study subjects where cytokine data are available. Do the authors have any information about the development of the neonates, for instance the infection rate during the first year of life? This could be correlated to the composition of the colostrum cytokines.

Answer: We agree with reviewer concern and clarify that we’ve characterized the population considering the period of the study, therefore, all patients were included, and the analysis of cytokines were restricted due to the assays - please see the answer to question #1, reviewer #1.

Although neonatal information would add valuable mean to the study, the main objective was to perform a cross-sectional study, to characterize the population and bring data regarding cytokines in colostrum at extremes of reproductive age, as this data are scarce. Moreover, we don’t have such data as this work was performed during a Master Course, with maximum of 2 years for collection, analysis and publication, so, unfortunately, the follow up was not possible.

5) Very often, it is not clear between which groups a significant difference was detected. Did the authors include those samples where cytokines could not be detected at all in their analysis?

Answer: We thanks for this observation, thus, changes were made along the text and in the Tables’ legends to make clear the significant differences. Moreover, we’ve included the description how samples were analyzed regarding the detection limit (lines 158-163).

What can the authors tell about changes in the colostrum cytokine composition within the first 72h after birth? These changes may influence their results as samples were taken between 48 and 72h after delivery.

Answer: We understand reviewer’s concern and we justify the interval 48-72h as colostrum secretion based on the literature (1-4). Although with some variation in time definition, mainly milk secretion consists of 3 phases: the first is colostrum produced in small quantities in the first five days after delivery (colostrum overload occurs 2 to 3 days after delivery) and is rich in cytokines, biochemical and immunological components; the second phase is the transition milk from the sixth day until the end of the second week after delivery, still with colostrum characteristics and the third stage is considered the mature milk that occurs from 15 days postpartum, with the highest composition of water, around 70% (1-4). Additionally, we took care of collecting half of the samples in the period of 48h postpartum, and half of them after 72h to avoid variations in the cytokines concentrations with time (we’ve included this detail in Material and Methods section, lines 141-143). Therefore, we believe time-collection had no influence in our results. 

1- “colostrum (once in the first 6 days of life) and mature human milk (once at 4–6 weeks postpartum” Munblit D, Treneva M, Peroni DG, Colicino S, Chow L, Dissanayeke S, et al. Colostrum and mature human milk of women from London, Moscow, and Verona: determinants of immune composition. Nutrients. 2016; 8(11). pii: E695. doi:10.3390/nu8110695. 

2- “colostrum (1st–6th day postpartum), transitional (7th–15th day postpartum) and mature milk (from the 16th day onwards)”

Collado MC, Santaella M, Mira-Pascual L, Martínez-Arias E, Khodayar-Pardo P, Ros G, et al. Longitudinal study of cytokine expression, lipid profile and neuronal growth factors in human breast milk from term and preterm deliveries. Nutrients 2015;7(10):8577-91. doi: 10.3390/nu7105415.

3- Pang WW, Hartmann PE. Initiation of human lactation: secretory differentiation and secretory activation. J Mammary Gland Biol Neoplasia. 2007;12: 211–21.

4- “colostrum was collected from 24-72 hours postpartum, and mature milk was collected at the end of the first month”

Freitas NA, Santiago LTC, Kurokawa CS, Meira Júnior JD, Corrente JE, Rugolo LMSS. Effect of preeclampsia on human milk cytokine levels. J Matern Fetal Neonatal Med.2018;25:1-5. doi: 10.1080/14767058.2018.1429395.

Reviewer #4: The current study compared the concentrations of IL-1β, IL-6, IL-8, and TNF-α cytokines in the colostrum of adolescent mothers and those with advanced maternal age and to describe sociodemographic, gestational, and perinatal characteristics. The large sample size and cross sectional nature of the study is a significant strength of the study. The main findings of the study are that concentrations of IL-1β and IL-6 were higher in the colostrum of mothers with advanced age compared to adolescent mothers, but did not differ from the control group. Although the findings are modest, the study is carefully conducted, dataset is rich with regards to prenatal and postnatal observations in the mothers and offspring, limitations are acknowledged and the manuscript is well written. I have some minor comments that need to be addressed below.

• What ELISA platform (multiplex, single plex, company) was used for the cytokine analysis?

Answer: We provided the detailed description of the assays in the revised version of the manuscript, as well as the technique protocol was sent for assigned its own identifier (DOI), as suggested by Plos One. 

Lines 158-165: “Commercially available ELISA kits (R&D Systems, Minneapolis, MN, USA) were used to quantify individually the IL-1β, IL-6, IL-8 e TNF-α levels in the studied samples. All assays were performed according to the manufacturer’s instructions. Samples in which cytokine levels were estimated to be below the sensitivity of the assay were set equal to the sensitivity of the assay and those with concentrations at levels above standard curve were diluted and re-assayed. The assays readers were performed in an ELISA automatic reader (Epoch-BioTek, Winooski, VT, USA), at a wavelength of 492 nm.”

---

## [Decision Letter · Decision Letter 1]

24 Mar 2020

PONE-D-19-35580R1

Profile of pro-inflammatory cytokines in colostrum of nursing mothers at the extremes of reproductive age

PLOS ONE

Dear Dr Morceli,

Thank you for submitting your manuscript to PLOS ONE. After careful consideration, we feel that it has merit but does not fully meet PLOS ONE’s publication criteria as it currently stands. Therefore, we invite you to submit a revised version of the manuscript that addresses ALL the points raised during the review process.

We would appreciate receiving your revised manuscript by May 08 2020 11:59PM. To enhance the reproducibility of your results, we recommend that if applicable you deposit your laboratory protocols in protocols.io, where a protocol can be assigned its own identifier (DOI) such that it can be cited independently in the future. For instructions see: http://journals.plos.org/plosone/s/submission-guidelines#loc-laboratory-protocols

We look forward to receiving your revised manuscript.

Kind regards,

Frank T. Spradley

Academic Editor

PLOS ONE

Reviewers' comments:

Reviewer's Responses to Questions

**Comments to the Author**

1. If the authors have adequately addressed your comments raised in a previous round of review and you feel that this manuscript is now acceptable for publication, you may indicate that here to bypass the “Comments to the Author” section, enter your conflict of interest statement in the “Confidential to Editor” section, and submit your "Accept" recommendation.

Reviewer #1: All comments have been addressed

Reviewer #2: All comments have been addressed

Reviewer #3: All comments have been addressed

Reviewer #4: All comments have been addressed

2. Is the manuscript technically sound, and do the data support the conclusions?

Reviewer #1: Yes

Reviewer #2: Yes

Reviewer #3: (No Response)

Reviewer #4: Yes

3. Has the statistical analysis been performed appropriately and rigorously? 

Reviewer #1: Yes

Reviewer #2: Yes

Reviewer #3: (No Response)

Reviewer #4: Yes

4. Have the authors made all data underlying the findings in their manuscript fully available?

Reviewer #1: Yes

Reviewer #2: Yes

Reviewer #3: (No Response)

Reviewer #4: Yes

5. Is the manuscript presented in an intelligible fashion and written in standard English?

Reviewer #1: Yes

Reviewer #2: Yes

Reviewer #3: (No Response)

Reviewer #4: Yes

6. Review Comments to the Author

Reviewer #1: Judged by the responses from the authors, most of the problems are resolved in the revised version. However, there remains some points for correction. These are given below.

<points>

1. Lines 41-42: “immunoenzymatic assay (ELISA)” should be changed to “enzyme-linked immunosorbent assay (ELISA)”.

2. Line 149: “72h” should be changed to “72 h”.

4. Line 154: “2mL” should be changed to “2 mL”.

5. Line 155: “160g” should be changed to “160 g”.

6. Line 189: “IL-8 e TNF-α” should be changed to “IL-8, and TNF-α”.

7. Lines 173-174: “Graph Pad Prism software, version 6.0” should be changed “Graph Pad Prism software, version 6.0 (San Diego, CA)”.

Reviewer #2: The question raised previously have been well addressed. There are a few typographic errors as indicated below:

Abstract: Line 41: ..attended at the.....should be replaced with: who attended the maternity-----

line 139:... should be punctuated as: advanced maternal age. This...

line 153: should be punctuated as: give birth. (included in the study group : should be deleted.)

Reviewer #3: (No Response)

Reviewer #4: (No Response)

  </points>

7. PLOS authors have the option to publish the peer review history of their article (what does this mean?). If published, this will include your full peer review and any attached files.

Reviewer #1: No

Reviewer #2: No

Reviewer #3: No

Reviewer #4: No

---

## [Author Response · Author response to Decision Letter 1]

30 Mar 2020

March 30 2020

Review: Manuscript number PONE-D-19-35580

Editorial Board, PLOS ONE

Thank you for the careful review of our work and for providing us the opportunity to submit a revised version of the manuscript that addresses the points below. 

All answers are detailed below each reviewers’ comments, as well highlighted in the revised version of the manuscript. 

REVIEW COMMENTS TO THE AUTHOR

Reviewer #1: Judged by the responses from the authors, most of the problems are resolved in the revised version. However, there remains some points for correction. 

Answer: We appreciate your careful review, and the suggested changes were performed, as highlighted in blue in the revised version of the manuscript in the respective lines named below.

1. Lines 41-42: “immunoenzymatic assay (ELISA)” should be changed to “enzyme-linked immunosorbent assay (ELISA)” (Line 41-42).

2. Line 149: “72h” should be changed to “72 h” (Line 149).

4. Line 154: “2mL” should be changed to “2 mL” (Line 154).

5. Line 155: “160g” should be changed to “160 g” (Line 155).

6. Line 189: “IL-8 e TNF-α” should be changed to “IL-8, and TNF-α” (Line 159).

7. Lines 173-174: “Graph Pad Prism software, version 6.0” should be changed “Graph Pad Prism software, version 6.0 (San Diego, CA)” (Line 173-174).

Reviewer #2: The question raised previously have been well addressed. There are a few typographic errors as indicated below:

Answer: We appreciate your careful review, and the suggested changes were performed, as highlighted in blue in the revised version of the manuscript in the respective lines named below.

1. Abstract: Line 41: attended at the.....should be replaced with: who attended the maternity—The study’s location had been suppressed from the abstract in the first revision, we include the suggestion grammar in the Methods section (Line 124)

2. line 139:... should be punctuated as: advanced maternal age. This...(Line 140)

3. line 153: should be punctuated as: give birth. (included in the study group : should be deleted.) 

We thank the reviewer and clarify that paragraph was rephrased in the revised version (Lines 139-141)

Hope these changes are satisfactory and will consider our paper for publication in PLoS ONE. 

Sincerely,

Morceli, G

---

## [Editor Report · Decision Letter 2]

3 Apr 2020

Profile of pro-inflammatory cytokines in colostrum of nursing mothers at the extremes of reproductive age

PONE-D-19-35580R2

Dear Dr. Morceli,

We are pleased to inform you that your manuscript has been judged scientifically suitable for publication and will be formally accepted for publication once it complies with all outstanding technical requirements.

With kind regards,

Frank T. Spradley

Academic Editor

PLOS ONE

---

## [Editor Report · Acceptance letter]

4 Jun 2020

PONE-D-19-35580R2 

Profile of pro-inflammatory cytokines in colostrum of nursing mothers at the extremes of reproductive age 

Dear Dr. Morceli:

I'm pleased to inform you that your manuscript has been deemed suitable for publication in PLOS ONE. Congratulations! Your manuscript is now with our production department. 

Kind regards, 

on behalf of

Dr. Frank T. Spradley 

Academic Editor

PLOS ONE